# A genome-to-genome analysis of associations between human genetic variation, HIV-1 sequence diversity, and viral control

István Bartha[1,2,3,4], Jonathan M Carlson[5†], Chanson J Brumme[6†], Paul J McLaren[1,2,4†], Zabrina L Brumme[6,7], Mina John[8], David W Haas[9], Javier Martinez-Picado[10,11], Judith Dalmau[10], Cecilio López-Galíndez[12], Concepción Casado[12], Andri Rauch[13], Huldrych F Günthard[14], Enos Bernasconi[15], Pietro Vernazza[16], Thomas Klimkait[17], Sabine Yerly[18], Stephen J O'Brien[19], Jennifer Listgarten[5], Nico Pfeifer[5‡], Christoph Lippert[5], Nicolo Fusi[5], Zoltán Kutalik[4,20], Todd M Allen[21], Viktor Müller[3], P Richard Harrigan[6,22], David Heckerman[5], Amalio Telenti[2*], Jacques Fellay[1,2,4*], for the HIV Genome-to-Genome Study and the Swiss HIV Cohort Study

[1]School of Life Sciences, École Polytechnique Fédérale de Lausanne, Lausanne, Switzerland; [2]Institute of Microbiology, University Hospital and University of Lausanne, Lausanne, Switzerland; [3]Research Group of Theoretical Biology and Evolutionary Ecology, Eötvös Loránd University and the Hungarian Academy of Sciences, Budapest, Hungary; [4]Swiss Institute of Bioinformatics, Lausanne, Switzerland; [5]eScience Group, Microsoft Research, Los Angeles, United States; [6]BC Centre for Excellence in HIV/AIDS, Vancouver, Canada; [7]Faculty of Health Sciences, Simon Fraser University, Burnaby, Canada; [8]Institute of Immunology and Infectious Diseases, Murdoch University, Murdoch, Australia; [9]Vanderbilt University Medical Center, Nashville, United States; [10]AIDS Research Institute IrsiCaixa, Institut d'Investigació en Ciències de la Salut Germans Trias i Pujol, Universitat Autònoma de Barcelona, Badalona, Spain; [11]Institució Catalana de Recerca i Estudis Avançats (ICREA), Barcelona, Spain; [12]Centro Nacional de Microbiología, Instituto de Salud Carlos III, Madrid, Spain; [13]Clinic of Infectious Diseases, University of Bern & Inselspital, Bern, Switzerland; [14]Division of Infectious Diseases and Hospital Epidemiology, University Hospital and University of Zürich, Zürich, Switzerland; [15]Division of Infectious Diseases, Regional Hospital of Lugano, Lugano, Switzerland; [16]Division of Infectious Diseases and Hospital Epidemiology, Cantonal Hospital, St. Gallen, Switzerland; [17]Department of Biomedicine, University of Basel, Basel, Switzerland; [18]Laboratory of Virology, Geneva University Hospitals, Geneva, Switzerland; [19]Theodosius Dobzhansky Center for Genome Bioinformatics, St. Petersburg State University, St. Petersburg, Russia; [20]Institute of Social and Preventive Medicine, University Hospital and University of Lausanne, Lausanne, Switzerland; [21]Ragon Institute of MGH, MIT, and Harvard, Massachusetts General Hospital, Boston, United States; [22]Faculty of Medicine, University of British Columbia, Vancouver, Canada

**\*For correspondence:** Amalio.
Telenti@chuv.ch (AT); jacques.
fellay@epfl.ch (JF)

†These authors contributed equally to this work

‡Present address: Department of Computational Biology and Applied Algorithmics, Max Planck Institute for Informatics, Saarbrücken, Germany

**Competing interests:** The authors declare that no competing interests exist.

**Reviewing editor**: Gil McVean, Oxford University, United Kingdom

**Abstract** HIV-1 sequence diversity is affected by selection pressures arising from host genomic factors. Using paired human and viral data from 1071 individuals, we ran >3000 genome-wide scans, testing for associations between host DNA polymorphisms, HIV-1 sequence variation and plasma viral load (VL), while considering human and viral population structure. We observed significant human SNP associations to a total of 48 HIV-1 amino acid variants (p<2.4 × 10$^{-12}$). All associated SNPs mapped to the HLA class I region. Clinical relevance of host and pathogen variation was assessed using VL results. We identified two critical advantages to the use of viral variation for identifying host factors: (1) association signals are much stronger for HIV-1 sequence variants than VL, reflecting the 'intermediate phenotype' nature of viral variation; (2) association testing can be run without any clinical data. The proposed genome-to-genome approach highlights sites of genomic conflict and is a strategy generally applicable to studies of host–pathogen interaction.

**eLife digest** Developing treatments or vaccines for HIV is challenging because the genetic makeup of the virus is constantly changing in an effort to outwit the human immune system. Moreover, the immune system is highly variable as a result of the long-standing co-evolution of humans and microbes. Each individual will try to oppose the invading virus in a unique way, forcing the virus to acquire specific mutations that can be interpreted as the genetic signature of this one-against-one battle.

To explore the influence of co-evolution on HIV, Bartha et al. took samples of both human and viral genomes from 1071 individuals infected with HIV, the AIDS virus, and used genotyping and sequencing technology to obtain a comprehensive description of the genetic variation in both. Computational techniques were then used to search for links between variants in the human DNA sequences and variants in the viral sequences.

The most common type of genetic variation found in the human genome is a single nucleotide polymorphism, or SNP for short: a SNP is produced when a single nucleotide – an A, C, G or T – is replaced by a different nucleotide. Bartha et al. found that SNPs within the human DNA sequences in their study were linked to variations in 48 amino acids in HIV. Moreover, all these SNPs were found within a group of genes known as the HLA (human leukocyte antigen) system, which encodes for proteins that play a vital role in the immune response. This work identified the areas of the human genome that put pressure on the AIDS virus, and the regions of the virus that serve to escape human control.

The approach developed by Bartha et al. allows the interactions between a microbe and a human host to be studied by looking at the genome of the microbe and the genome of the infected person. It also differentiates host-induced mutations that limit the capacity of the virus to do harm from those that are tolerated by the pathogen. A similar strategy could be used to study other infectious diseases.

## Introduction

Through multiple rounds of selection and escape, host and pathogen genomes are imprinted with signatures of co-evolution that are governed by Darwinian forces. On the host side, well-characterized anti-retroviral restriction factors, such as *TRIM5α*, *APOBEC3G* and *BST2*, harbor strong signals of selection in primate genomes, clear examples of retroviral pressure (*Ortiz et al., 2009*). On the virus side, obvious signs of selection are observable in the HIV-1 genome: escape mutations and reversions have been described in epitopes restricted by human leukocyte antigen (HLA) class I molecules and targeted by cytotoxic T lymphocyte (CTL) responses (*Goulder et al., 2001*; *Kawashima et al., 2009*). Sequence polymorphisms have also been reported recently in regions targeted by killer immunoglobulin-like receptors (KIR), suggesting evasion from immune pressure by natural killer (NK) cells (*Alter et al., 2011*). Evidence for the remodeling of retroviral genomes by host genetic pressure also comes from simian immunodeficiency virus (SIV) infection studies in rhesus macaques, where escape from restrictive *TRIM5α* alleles has been observed in the viral capsid upon cross-species transmission of SIVsm

(*Kirmaier et al., 2010*). In contrast, human alleles of *TRIM5α* do not result in escape mutations, likely because of adaptation of the pathogen to the host (*Rahm et al., 2013*). Sequence adaptation is also a known feature of cross-species transmission. For example, a methionine in the matrix protein (Gag-30) in SIV$_{cpzPtt}$ changed to arginine in lineages leading to HIV-1 and reverted to methionine when HIV-1 was passaged through chimpanzees (*Wain et al., 2007*).

To date, combined analyses of human and HIV-1 genetic data have addressed the association of HLA and KIR genes with variants in the retroviral genome (*Moore et al., 2002*; *Brumme et al., 2007*; *Bhattacharya et al., 2007*; *Kawashima et al., 2009*; *Alter et al., 2011*; *Carlson et al., 2012*; *Wright et al., 2012*). Additionally, genome-wide association studies (GWAS) performed in the host have focused on various HIV-related clinical phenotypes (*Fellay et al., 2007*; *Fellay et al., 2009*; *Pereyra et al., 2010*). In parallel, large amounts of HIV-1 sequence data have been generated for phylogenetic studies, which shed new light on viral transmission and evolution (*Kouyos et al., 2010*; *Alizon et al., 2010*; *Von Wyl et al., 2011*), or allow clinically driven analyses of viral genes targeted by antiretroviral drugs (resistance testing) (*Von Wyl et al., 2009*).

Building on the unprecedented possibility to acquire and combine paired human and viral genomic information from the same infected individuals; we employ an innovative strategy for global genome-to-genome host–pathogen analysis. By simultaneously testing for associations between genome-wide human variation, HIV-1 sequence diversity, and plasma viral load (VL), our approach allows the mapping of all sites of host–pathogen genomic interaction, the correction for both host and viral population stratification, and the assessment of the respective impact of human and HIV-1 variation on a clinical outcome (*Figure 1*).

## Results

### Study participants, host genotypes, and HIV-1 sequence variation

Full-length HIV-1 genome sequence and human genome-wide SNP data were obtained from seven studies or institutions on a total of 1071 antiretroviral naive patients of Western European ancestry, infected with HIV-1 subtype B. The homogeneity of the study population was confirmed by principal component analysis of the genotype matrix: together, the first five principal components explained 1% of total genotypic variation. After quality control of the human genotype data, imputation and filtering, ~7 million SNPs were available for association testing. The full-length HIV-1 sequence is approximately 9.5 Kb long, corresponding to over 3000 encoded amino acids. Not all sequences were complete; on an average, viral residues were covered in 85% of the study population (range: 75% in Tat to 95% in Gag). Due to its hypervariable nature, the portion of the HIV-1 envelope gene that encodes the gp120 protein was not sequenced in most study samples and was therefore excluded. Overall 1126 residues of the HIV-1 proteome were found to be variable in at least 10 samples, for a total of 3381 different viral amino acids that could be represented by 3007 distinct binary variables.

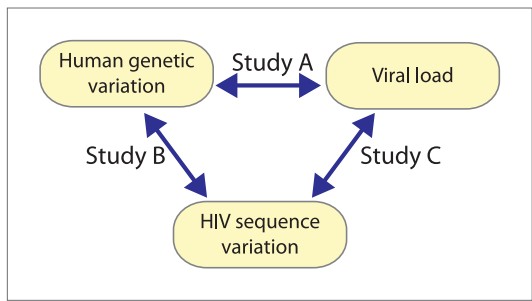

**Figure 1**. A triangle of association testing. The following association analyses were performed: [Study A] human SNPs vs plasma viral load (1 GWAS); [Study B] human SNPs vs variable HIV-1 amino acids (3007 GWAS); and [Study C] variable HIV-1 amino acids vs plasma viral load (1 proteome-wide association study).

### Host VL GWAS

We first performed a classical GWAS of host determinants of HIV-1 VL (*Figure 2A*, Study A) using data from 698 patients (65% of the study population) for whom a VL phenotype could be reliably estimated. The top associations were observed in the HLA class I region on chromosome 6 and were highly consistent with results observed previously (*Fellay et al., 2009*; *Pereyra et al., 2010*). The strongest associated SNP, rs9267454 (p = 1.5 × 10$^{-8}$), is in partial linkage disequilibrium (LD) with HLA-B*57:01 (r$^2$ = 0.47, D′ = 0.92), HLA-B*14:01 (r$^2$ = 0.12, D′ = 1.0), HLA-B*27:05 (r$^2$ = 0.01, D′ = 0.99), and the HLA-C -35 rs9264942 SNP (r$^2$ = 0.07, D′ = 0.77), and thus reflects these well-known associations with HIV-1 control. These results confirm the quality of the study population for the purpose of genome analysis of determinants of HIV-1-related outcomes.

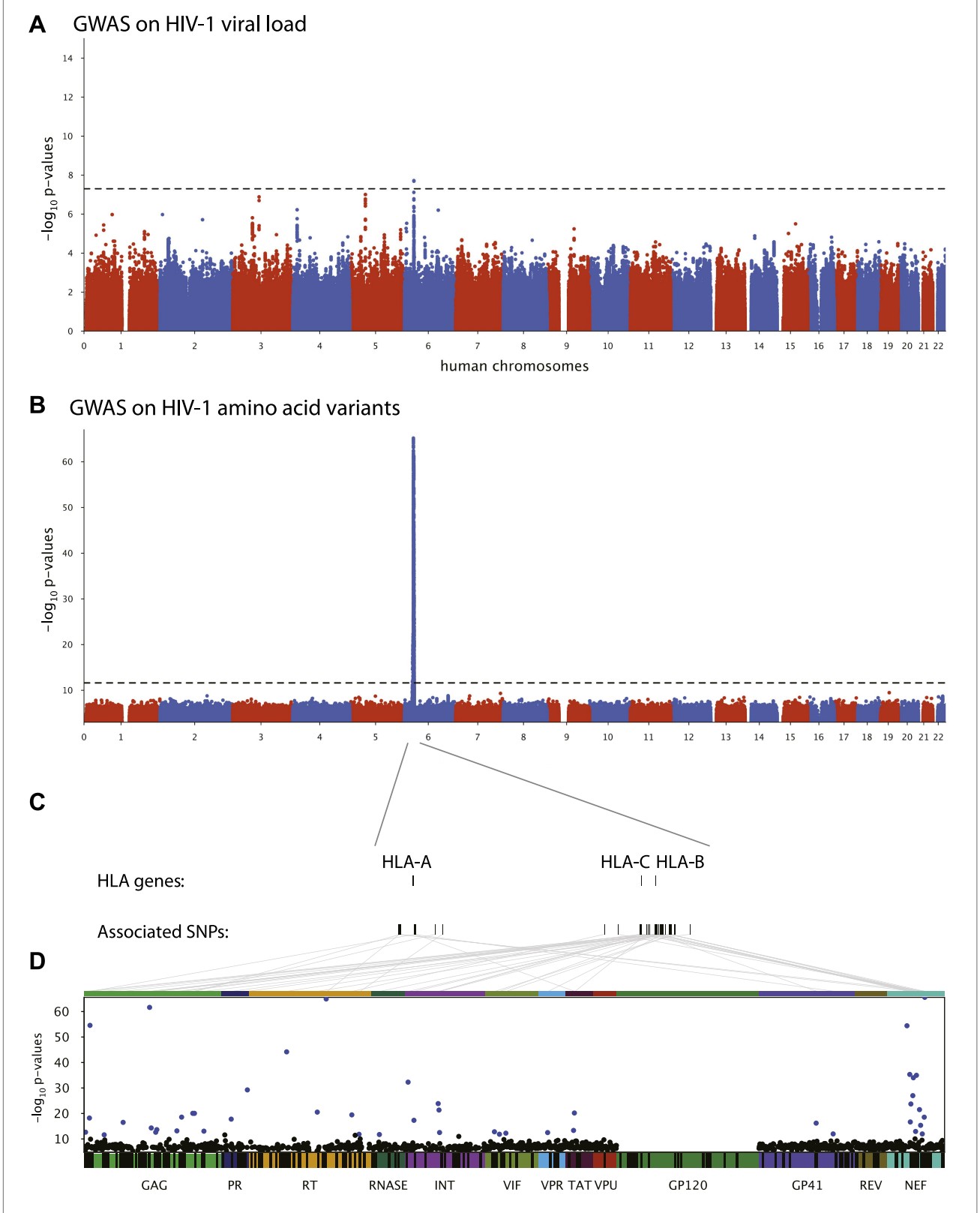

**Figure 2**. Results of the genome-wide association analyses. (**A**) Associations between human SNPs and HIV-1 plasma viral load. The dotted line shows the Bonferroni-corrected significance threshold (p-value < 7.25 × 10⁻⁹). (**B**) Associations between human SNPs and HIV-1 amino acid variants, with 3007 GWAS collapsed in a single Manhattan plot. The dotted line shows the Bonferroni-corrected significance threshold (p-value < 2.4 × 10⁻¹²). (**C**) Schematic representation

*Figure 2. Continued on next page*

*Figure 2. Continued*

of the HLA class I genes and of the SNPs associated with HIV-1 amino acid variants in the region. (**D**) Same association results as in panel **B**, projected on the HIV-1 proteome. Only the strongest association is shown for each amino acid. Significant associations are indicated by a blue dot. The gp120 part of the HIV-1 proteome was not tested. The colored bar below the plot area shows the positions of the optimally defined CD8+ T cell epitopes. An interactive version of this figure can be found at http://g2g.labtelenti.org (which is also available to download from Zenodo, http://dx.doi.org/10.5281/zenodo.7138).

## Genome-to-genome analyses

3007 genome-wide analyses of associations between human SNPs and HIV-1 amino acid variants were performed in the full sample of 1071 individuals (*Figure 2B*, <u>Study B</u>) using logistic regression corrected for viral phylogeny (*Carlson et al., 2008*; *Carlson et al., 2012*). Highly significant associations were observed between SNPs in the major histocompatibility complex (MHC) region and multiple amino acids throughout the HIV-1 proteome (except in Vpu, Rev and the RNaseH subunit of RT) (*Figure 2C*), with Gag and Nef having a significantly higher density of associated variable sites than the rest of the proteome (Gag: 6.8% vs 2.6% p=0.001; Nef: 11% vs 2.6% p = $1.2 \times 10^{-5}$, binomial tests). Using Bonferroni correction for multiple testing (threshold p = $2.4 \times 10^{-12}$), significant human SNP associations were observed with 48 viral amino acids (*Figure 2* and *Table 1*). None of these 48 amino acids mapped to known sites of major antiretroviral drug resistance mutations (*Hirsch et al., 2008*). The strongest association found was between rs72845950 and Nef position 135 (p = $2.7 \times 10^{-66}$). Associations were much stronger between human SNPs and HIV-1 amino acids than with VL. For example, the SNP rs2395029, a proxy for HLA-B*57:01 ($r^2 = 0.93$), has a p-value of $1.21 \times 10^{-6}$ for association with VL, while it reaches a p-value of $4 \times 10^{-59}$ for association with amino acid variation in Gag at position 242 (a well known position of escape from HLA-B*57:01). No significant signals were identified outside the MHC. A link to the complete set of association results can be found at http://g2g.labtelenti.org (which is also available to download from Zenodo, http://dx.doi.org/10.5281/zenodo.7139). These results demonstrate the feasibility and improved power of performing association testing using viral genetic variation as outcome, independent of clinical phenotype.

## SNPs, HLA alleles and CTL epitopes

We next assessed whether the top SNPs associated with HIV-1 amino acids represent indirect markers of HLA class I alleles known to exert evolutionary pressure on HIV-1 (*Table 1*). We tested pairwise correlations between significant MHC SNPs and HLA class I alleles. The analysis confirmed the existence of high LD between SNPs and HLA alleles targeting corresponding epitopes. For example, the strongest association (p = $2.7 \times 10^{-66}$) was observed between residue 135 in Nef, located in an optimally defined A*24:02 epitope, and rs72845950, which strongly tags HLA-A*24:02 ($r^2 = 0.89$). Furthermore, we observed that a substantial fraction of the identified viral amino acids (24/48, 50%) were located within an optimally defined CTL epitope restricted by one or more HLA alleles tagged by the associated SNP (http://www.hiv.lanl.gov/content/immunology/tables/optimal_ctl_summary.html, supplemented with a recently updated list of epitopes [*Carlson et al., 2012*]). However, in seven cases, the classical HLA allele implicated through LD with a tagging SNP did not match previously reported restriction patterns (*Table 1*). These data demonstrate that this approach can reconstruct a map of targets of HLA pressure across the viral proteome and identify sites outside classical epitopes that could represent additional escape variants or compensatory mutations. That a substantial proportion of associated viral amino acids lay outside known CTL epitopes also highlights this approach as a tool to guide novel epitope discovery (*Bhattacharya et al., 2007*; *Almeida et al., 2011*).

Analysis of polymorphic amino acids within the HLA genes has been shown to improve power for detection of association with clinical outcome and has demonstrated the biological relevance of key residues in the HLA-B binding groove (*Pereyra et al., 2010*). Therefore, we used the genome-to-genome framework to characterize the evolutionary pressure of HLA class I amino acids on the viral genome. The top associations in all classical class I genes mapped to discrete residues in the binding grooves of the HLA molecule: HLA-A position 62 (p = $3.3 \times 10^{-76}$ with HIV Nef 135), HLA-B position 70 (p = $7.1 \times 10^{-57}$ with HIV Gag 242), HLA-C position 99 (p = $5.4 \times 10^{-63}$ with HIV Nef 70). These data indicate that all class I HLA genes can exert strong pressure on the viral proteome through a shared mechanism. The association results for HLA amino acids can also be found at http://g2g.labtelenti.org (which is also available to download from Zenodo).

**Table 1.** Associations between HIV-1 amino acid variants and human polymorphisms

| HIV gene | HIV position | SNP | CTL epitope (codons) | Tagging HLA (D'/r²) | SNP vs aa (p) | SNP vs VL (p) | aa vs VL (p) |
|---|---|---|---|---|---|---|---|
| GAG | 12 | chr6:31285512 | – | B*49:01 (1.00/1.00) | 2.20E-13 | 6.70E-01 | 5.60E-01 |
| GAG | 26 | rs12524487 | – | B*15:01 (1.00/0.82) | 6.10E-19 | 2.10E-01 | 1.40E-01 |
| GAG | 28 | rs1655912 | RLRPGGKKK (20–28) | A*03:01 (1.00/0.81) | 2.70E-55 | 5.60E-01 | 2.00E-02 |
| GAG | 79 | chr6:31267544 | LYNTVATL (78-85) | C*14:02 (1.00/0.96) | 2.40E-12 | 3.50E-01 | 2.80E-01 |
| GAG | 147 | rs1055821 | – | C*06:02 (0.95/0.71) | 3.10E-17 | 3.30E-07 | 2.90E-05 |
| GAG | 242 | rs73392116 | TSTLQEQIGW (240–249) | B*57:01 (1.00/0.98) | 2.40E-62 | 1.90E-06 | 1.70E-05 |
| GAG | 248 | rs41557213 | TSTLQEQIGW (240–249) | B*57:01 (1.00/0.97) | 4.80E-15 | 2.00E-06 | 5.30E-03 |
| GAG | 264 | chr6:31376564 | KRWIILGLNK (263–272) | B*27:05 (1.00/0.92) | 2.30E-13 | 5.50E-02 | 3.50E-01 |
| GAG | 268 | rs2249935 | GEIYKRWIIL (259–268) | B*08:01 (1.00/0.43) | 2.20E-14 | 5.10E-01 | 1.90E-01 |
| GAG | 340 | rs11966319 | – | B*15:01 (0.94/0.42) | 6.70E-14 | 4.60E-01 | 7.70E-01 |
|  |  |  |  | C*03:04 (0.99/0.59) |  |  |  |
| GAG | 357 | rs2523612 | GPGHKARVL (355–363) | B*07:02 (0.99/0.95) | 2.70E-19 | 2.20E-01 | 1.20E-01 |
|  |  |  |  | C*07:02 (0.99/0.84) |  |  |  |
| GAG | 397 | rs61754472 | – | A*31:01 (0.97/0.83) | 8.80E-21 | 3.50E-01 | 8.30E-01 |
| GAG | 403 | rs28896571 | – | – | 8.90E-21 | 7.90E-01 | 8.60E-01 |
| GAG | 437 | rs34268928 | RQANFLGKI (429-437) | B*13:02 (1.00/0.96) | 8.70E-14 | 1.80E-02 | 6.80E-02 |
| GP41 | 206 | rs17881210 | – | B*15:01 (1.00/0.88) | 6.10E-17 | 6.10E-01 | 3.00E-01 |
| GP41 | 267 | rs9278477 | RLRDLLLIVTR (259–269) | A*03:01 (1.00/0.01) | 1.00E-12 | 7.80E-01 | 2.60E-01 |
| INT | 11 | rs2596477 | – | B*44:02 (1.00/0.64) | 5.10E-33 | 1.50E-01 | 1.80E-01 |
| INT | 32 | rs1050502 | – | B*51:01 (0.97/0.92) | 4.80E-18 | 7.20E-01 | 4.00E-01 |
| INT | 119 | rs9264954 | – | C*05:01 (1.00/1.00) | 1.30E-24 | 7.90E-01 | 1.10E-01 |
| INT | 122 | rs9264419 | – | C*05:01 (1.00/0.95) | 4.50E-22 | 8.30E-01 | 7.80E-01 |
| INT | 124 | chr6:31345421 | STTVKAACWW (123–132) | B*57:01 (1.00/1.00) | 3.00E-13 | 1.10E-06 | 9.70E-03 |
| NEF | 71 | rs2596488 | FPVTPQVPLR (68–77) | B*07:02 (1.00/0.98) | 3.80E-55 | 2.50E-01 | 8.10E-02 |
|  |  |  | – | C*07:02 (0.95/0.83) |  |  |  |

*Table 1. Continued on next page*

*Table 1. Continued*

| HIV gene | HIV position | SNP | CTL epitope (codons) | Tagging HLA (D′/r²) | SNP vs aa (p) | SNP vs VL (p) | aa vs VL (p) |
|---|---|---|---|---|---|---|---|
| NEF | 81 | rs9295987 | RPMTYKAAL (77–85) | B*07:02 (1.00/0.01) | 4.80E-36 | 2.50E-01 | 9.50E-02 |
| | | | – | C*04:01 (0.90/0.63) | | | |
| NEF | 83 | rs34768512 | – | B*15:01 (1.00/0.47) | 2.20E-17 | 2.80E-01 | 1.50E-02 |
| | | | – | C*03:04 (0.96/0.54) | | | |
| NEF | 85 | rs2395475 | RPMTYKAAL (77–85) | B*07:02 (1.00/0.29) | 1.90E-24 | 8.10E-01 | 1.30E-03 |
| | | | – | B*08:01 (1.00/0.22) | | | |
| | | | – | C*07:02 (0.97/0.30) | | | |
| NEF | 92 | rs16896166 | AVDLSHFLK (84–92) | A*11:01 (1.00/0.99) | 1.00E-27 | 5.30E-01 | 1.50E-01 |
| NEF | 94 | rs9265972 | FLKEKGGL (90–97) | B*08:01 (1.00/0.97) | 9.60E-35 | 9.80E-01 | 1.20E-01 |
| NEF | 102 | rs2524277 | – | B*44:03 (0.98/0.96) | 1.10E-13 | 4.40E-01 | 2.40E-01 |
| NEF | 105 | rs1049709 | – | C*07:01 (1.00/0.98) | 1.10E-35 | 9.00E-01 | 2.70E-01 |
| NEF | 116 | chr6:31402358 | HTQGYFPDW (116–124) | B*57:01 (1.00/1.00) | 3.00E-22 | 1.90E-06 | 3.30E-01 |
| NEF | 120 | chr6:31236168 | - | C*14:02 (1.00/1.00) | 4.40E-16 | 3.60E-01 | 1.20E-02 |
| NEF | 126 | chr6:31102273 | – | B*51:01 (1.00/0.18) | 1.10E-12 | 1.80E-01 | 4.90E-02 |
| NEF | 133 | chr6:31397689 | – | B*35:01 (0.95/0.89) | 2.80E-19 | 2.50E-01 | 3.40E-01 |
| NEF | 135 | rs72845950 | RYPLTFGW (134–141) | A*24:02 (1.00/0.88) | 2.70E-66 | 9.10E-02 | 5.50E-03 |
| PR | 35 | rs2523577 | EEMNLPGRW (34–42) | B*44:02 (1.00/0.64) | 1.70E-18 | 1.60E-01 | 5.70E-01 |
| PR | 93 | rs2263323 | – | B*15:01 (0.98/0.92) | 5.60E-30 | 4.70E-01 | 9.50E-01 |
| RNASE | 28 | rs2428481 | – | B*08:01 (1.00/1.00) | 1.80E-12 | 8.10E-01 | 6.20E-01 |
| RT | 135 | rs1050502 | TAFTIPSI (128–135) | B*51:01 (0.97/0.92) | 6.70E-45 | 7.20E-01 | 3.00E-01 |
| RT | 245 | chr6:31411714 | IVLPEKDSW (244–252) | B*57:01 (1.00/0.98) | 2.90E-21 | 1.20E-06 | 5.40E-02 |
| RT | 277 | rs3128902 | QIYPGIKVR (269–277) | A*03:01 (1.00/0.99) | 1.20E-65 | 8.20E-01 | 2.70E-01 |
| RT | 369 | rs17190134 | – | B*13:02 (0.93/0.86) | 3.50E-20 | 6.40E-02 | 1.40E-01 |
| RT | 395 | rs17194293 | – | - | 1.50E-12 | 1.20E-01 | 7.70E-02 |

*Table 1. Continued on next page*

Table 1. Continued

| HIV gene | HIV position | SNP | CTL epitope (codons) | Tagging HLA (D'/r²) | SNP vs aa (p) | SNP vs VL (p) | aa vs VL (p) |
|---|---|---|---|---|---|---|---|
| TAT | 29 | rs9260615 | – | A*32:01 (0.98/0.95) | 4.40E-14 | 3.90E-01 | 1.40E-01 |
| TAT | 32 | rs16899214 | CCFHCQVC (30–37) | C*12:03 (0.98/0.96) | 6.40E-21 | 3.40E-01 | 4.90E-01 |
| VIF | 33 | chr6:31430060 | ISKKAKGWF (31–39) | B*57:01 (1.00/0.98) | 1.50E-13 | 9.90E-07 | 9.30E-03 |
| VIF | 51 | rs7767850 | – | B*49:01 (1.00/1.00) | 1.40E-12 | 5.20E-01 | 2.10E-01 |
| VIF | 74 | rs2395029 | – | B*57:01 (1.00/0.98) | 5.40E-13 | 9.70E-07 | 2.80E-01 |
| VPR | 32 | chr6:31362941 | VRHFPRIWL (31–39) | B*27:05 (1.00/0.94) | 3.10E-13 | 5.40E-02 | 6.50E-01 |

Significant associations (p < 2.4 × 10⁻¹²) were observed for 48 HIV-1 amino acid variants. The table shows the major amino acid variants present at each specific HIV-1 position, the strongest associated SNP and its linked HLA class I allele(s), if applicable. The column 'CTL Epitope (codons)' lists published, optimally described CTL epitopes (available at http://www.hiv.lanl.gov/content/immunology/tables/optimal_ctl_summary.html and in [**Carlson et al., 2012**]) restricted by the tagged HLA class I allele(s) specified, and their positions within the protein. Where multiple overlapping epitopes restricted by the same HLA class I allele have been described, only one is shown. Associations where no relevant CTL epitope has been described are indicated with a dash. The last three columns give association p-values for comparisons between human SNPs and viral amino acids, human SNPs and plasma VL and viral amino acids and plasma VL, respectively. For tests involving viral amino acids accommodating more than 1 alternate allele, the smallest association p-value observed at that position is reported.

### HIV-1 amino acids vs plasma viral load

To address whether there was an observable impact of viral mutation on a clinical outcome in this sample, we tested for associations between all HIV-1 amino acid variant and VL (Study C). After correction for multiple testing (p threshold = $1.6 \times 10^{-5}$ based on 3125 viral amino acids), we did not observe any significant associations. We then focused on estimating the changes in VL associated with the 48 HIV-1 amino acid variants that were identified as significantly associated with host SNPs (*Figure 2D*). The effects of amino acid variation at these sites on VL ranged from –0.16 to +0.07 $\log_{10}$ copies/ml (*Figure 3A*). We also explored the combined fitness effect of multiple HIV-1 viral amino acid variants targeted by a single host marker using the well-understood model of HLA-B*57:01. We evaluated the effect on VL of 23 viral residues that associated with host variant rs2395029 ($r^2 = 0.93$ with

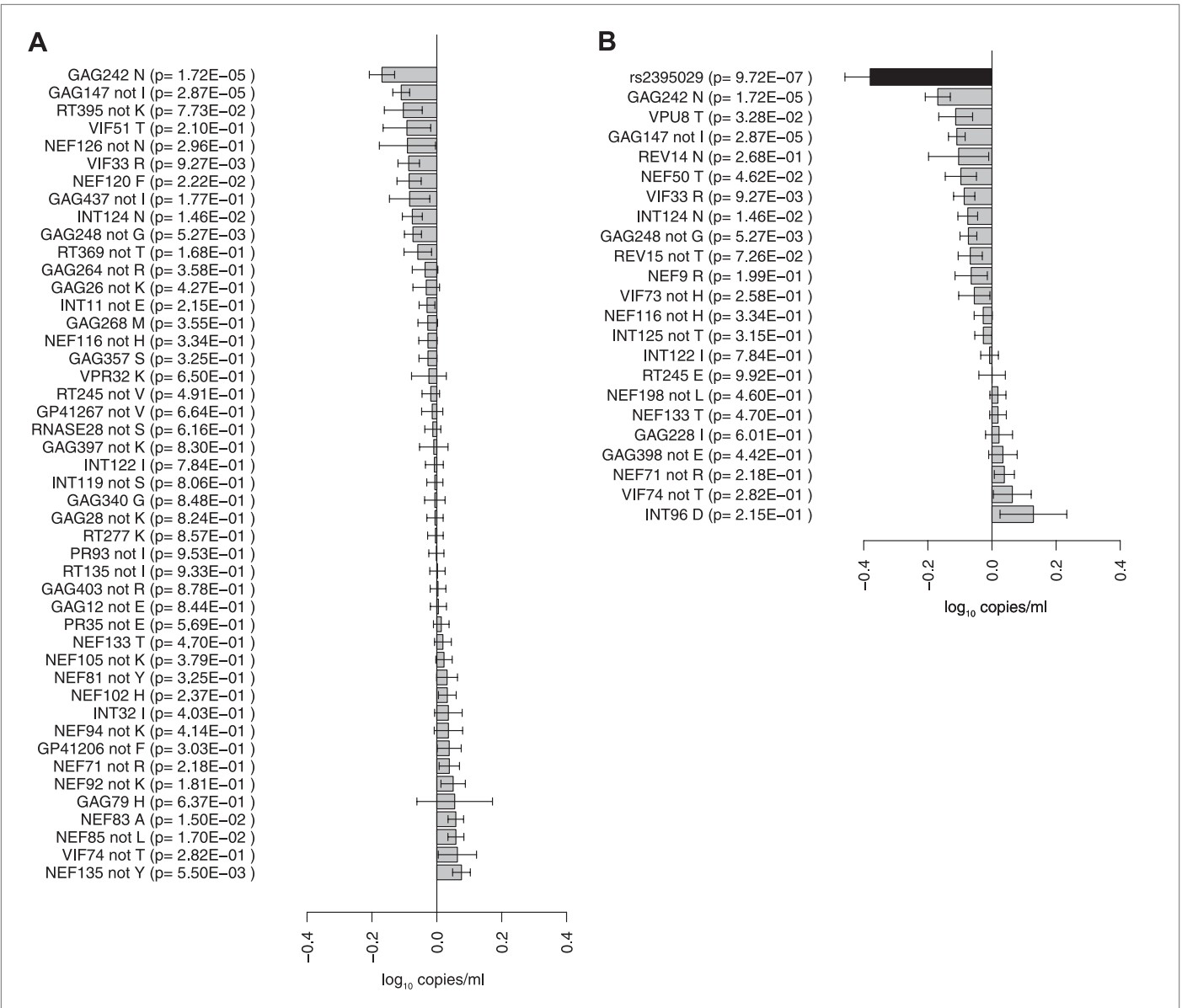

**Figure 3**. Association of HIV-1 amino acid variants with plasma viral load. (A) Changes in VL (slope coefficients from the univariate regression model and standard error, $\log_{10}$ copies/ml) for the 48 HIV-1 amino acids that are associated with host SNPs in the genome-to-genome analysis. (**B**) rs2395029, a marker of HLA-B*57:01 is associated with a 0.38 $\log_{10}$ copies/ml lower VL (black bar) in comparison to the population mean. Gray bars represent changes in VL for amino acid variants associated with rs2395029 (p<0.001). In case of multiallelic positions, the change in VL is shown for all minor amino acids combined vs the major amino acid (e.g., GAG147 not I).

HLA-B*57:01) in the genome-to-genome analysis (selection cutoff: p<0.001). The marker rs2395029 was associated with a 0.38 log decrease in viral RNA copies/ml. The univariate effect on VL for each of the 23 viral amino acids targeted by this allele ranged from –0.16 to +0.12 (*Figure 3B*). These results suggest that the genome-to-genome approach can be linked to clinical/laboratory phenotypes, allowing for detailed understanding of the distribution and relative contribution of sites of host–pathogen interaction to disease outcome.

## Discussion

HIV-1 host genomic studies performed so far have focused on clinically defined outcomes (resistance to infection, clinical presentation, disease progression or death) or on pathogen-related laboratory results (such as CD4+ T cell counts and VL set point). While useful, these phenotypes have significant drawbacks. First, consistency of phenotypic determination can be hard to achieve, and such inconsistency can adversely affect power in large-scale genetic studies performed across multiple centers (*Evangelou et al., 2011*). Second, a relatively long follow-up in the absence of antiretroviral treatment is necessary to obtain informative data about the natural history of infection. However, international guidelines now propose an early start of antiretroviral therapy in most HIV-1 infected individuals (*Thompson et al., 2012*), making the collection of large numbers of long-term untreated patients not only unrealistic but also ethically questionable.

To overcome these limitations, we developed a novel approach for host genetic studies of infectious diseases, built on the unprecedented possibility to obtain paired genome-wide information from hosts and pathogens. We combined human polymorphism and HIV-1 sequence diversity in the same analytical framework to search for sites of human-virus genomic conflict, effectively using variation in HIV-1 amino acids as an 'intermediate phenotype' for association studies. Intermediate phenotypes have recently been shown to be useful in uncovering association signals that are not detectable using more complex clinical endpoints: illustrative examples include metabolomic biomarkers in cardiovascular research (*Suhre et al., 2011*), serum IgE concentration in the study of asthma (*Moffatt et al., 2010*), or neuroimaging-based phenotypes in psychiatry genetics (*Rasetti and Weinberger, 2011*). Variation in the pathogen sequence is an as-yet-untapped intermediate phenotype, specific by nature to genomic research in infectious diseases. Importantly, it depends on sequencing the pathogen, which could prove in many cases easier and more standardized than obtaining detailed clinical phenotypes.

Our approach allowed the mapping of host genetic pressure on the HIV-1 genome. The strongest association signals genome-wide were observed between human SNPs tagging HLA class I alleles and viral mutations in their corresponding CTL epitopes. Additional association signals were observed outside of optimally defined CTL epitopes, which could indicate novel epitopes, or represent secondary (compensatory) mutations. In a single experiment, these results recapitulate extensive epidemiological and immunogenetic research and represent a proof-of-concept that biologically meaningful association signals are identifiable using a hypothesis-free strategy. Indeed, host factors leading to viral adaptation can be uncovered by searching for associated imprints in the viral genome. Of note, the International HIV Controllers Study demonstrated the importance of specific amino acid positions in the HLA-B binding groove on a clinical outcome (elite control) (*Pereyra et al., 2010*). We here extend this observation to the HLA-A and C grooves, emphasizing the similarity in mechanism of host pressure on the viral proteome that is not necessarily translated into observable clinical outcomes.

We found a higher density of amino acid positions under selection in Gag and Nef compared with the rest of the HIV proteome. This is consistent with earlier findings that indicate the importance of Gag p24-specific CTL responses in slower progression to AIDS (*Borghans et al., 2007*; *Brennan et al., 2012*) or controller status (*Dyer et al., 2008*). Moreover, this further demonstrates that mapping host pressure on the pathogen proteome can reveal biologically relevant effects.

Analyses were performed using samples from clinically well-characterized patients, most of them with repeated and reliable HIV-1 VL measurements in the absence of antiretroviral therapy. We were thus able to compare the results of GWAS assessing human genetic determinants of mean VL, a standard clinical correlate of HIV-1 control, and genome-to-genome GWAS on amino acid variants in the viral proteome. The use of HIV-1 variation as outcome resulted in a considerable gain in power to detect host factors: the lowest p-values were observed for SNPs mapping to the HLA class I region in both approaches, but associations were much stronger with HIV-1 amino acid variation

than for HIV-1 VL ($2.7 \times 10^{-66}$ vs $1 \times 10^{-08}$), even when accounting for the increased number of multiple tests.

In addition to identifying sites of interaction between the host and the pathogen, the study design allowed the scoring of biological consequences of such interaction, by assessing associations between host-driven escape at viral sites and an in vivo phenotype (VL). For example, we decomposed the effect of rs2395029 (a marker of HLA-B*57:01) on VL to the effects of the multiple viral amino acid variants that are associated with that SNP. While some HIV-1 amino acid changes individually associate with decrease in VL, the compound image that emerges is one of a multiplicity of modest effects distributed across many residues. Correlations between host-associated variants and VL are difficult to interpret, because they may reflect fitness costs or compensation, the existence of strong (*Iversen et al., 2006*; *Carlson et al., 2012*) or novel (*Almeida et al., 2011*) immune responses, or the indirect impact of specific HLA class I alleles. Nevertheless, the observation that the majority of host-associated HIV-1 mutations do not correlate with any detectable change in VL confirms HIV's remarkable capacity to adapt and compensate to immune pressure, often without measurable fitness cost.

A significant confounder in both human and viral genomic analyses is the existence of population stratification, where shared ancestry between infected individuals, stratification by ethnic groups, non-random distribution of HIV-1 subtypes, or clusters of viral transmission can all have an influence on the population frequencies of specific mutations, and thus create spurious associations if not carefully controlled for. Previous studies usually controlled for viral population substructure but were limited in the control of human population stratification (*Moore et al., 2002*; *Bhattacharya et al., 2007*). Our approach offers the opportunity to correct for both factors, thanks to the availability of extensive host and viral genomic information.

The present sample size provided approximately 80% power to detect a common human variant (minor allele frequency of 10%) with an odds ratio of 4.2 in the genome-to-genome analysis (Study B) and a viral amino acid explaining approximately 4% of the variation in plasma viral load (Study C) at the respective significance thresholds (*Purcell et al., 2003*). Consistent with most studies performed in HIV-1 host genetics over the past few years (reviewed in *Telenti and Johnson (2012)*), we did not identify previously unknown host genetic loci involved in host-viral interaction and HIV-1 restriction. The proposed approach can only detect polymorphic host factors that leave an imprint on the virus, which may exclude mediators of immunopathogenesis or genes involved in the establishment of tolerance (*Medzhitov et al., 2012*). An additional limitation is the incomplete nature of genomic information available both on the host side (common genotypes from GWAS) and on the viral side (near full-length consensus sequence; gp120 was not included in the analyses). Finally, the multiple hypothesis burden of a genome-to-genome scan is extremely high. It is conceivable that larger studies, or studies that focus on a subgroup of predefined host genes, would have power to detect novel associations. A comprehensive, but computationally challenging description of host–pathogen genomic interactions would require human genome sequencing, coupled with deep sequencing of intra-host retroviral subpopulations.

In summary, we used a genome-to-genome, hypothesis-free approach to identify associations between host polymorphisms and HIV-1 genomic variation. This strategy allows a global assessment of host–pathogen interactions at the genome level and reveals sites of genomic conflict. Comparable approaches are immediately applicable to explore other important infectious diseases, as long as polymorphic host factors exert sufficient selective pressure to trigger escape mutations in the pathogen. The observation that pathogen sequence variation, used as an intermediate phenotype, is more powerful than clinical and laboratory outcomes to identify some host factors allows smaller-scale studies and encourages analyses of less prevalent infectious diseases. Researchers involved in pathogen genome studies and host genetic studies should strongly consider the gathering of paired host–pathogen data.

## Materials and methods

### Ethics statement

Participating centers provided local Institutional Review Board approval for genetic analysis. Study participants provided informed consent for genetic testing, with the exception of a subset where a procedure approved by the relevant Research Ethics Board allowed the use of anonymized historical specimens in the absence of a specific informed consent.

## Participants

Study participants are treatment-naïve individuals followed in one of the following cohorts or institutions: the Swiss HIV Cohort Study (SHCS, www.shcs.ch, [*Schoeni-Affolter et al., 2010*]); the HAART Observational Medical Evaluation and Research (HOMER) study in Vancouver, Canada (www.cfenet.ubc.ca/our-work/initiatives/homer); the AIDS Clinical Trials Group (ACTG) Network in the USA (actgnetwork.org); the International HIV Controllers Study in Boston, USA (IHCS, www.hivcontrollers.org); Western Australian HIV Cohort Study, Perth, Australia; the AIDS Research Institute IrsiCaixa in Badalona, Spain; and the Instituto de Salud Carlos III in Madrid, Spain. To reduce noise due to host and viral diversity, we only included individuals of recent Western European ancestry (confirmed by clustering with HapMap CEU individuals in principal component analysis of the genotype data [*Price et al., 2006*]), and infected with HIV-1 subtype B (as assessed by the REGA Subtyping Tool [*De Oliveira et al., 2005*]). Plasma VL determinations in the absence of antiretroviral therapy were available from patients from the SHCS and the HOMER study. The VL phenotype was defined as the average of the $log_{10}$-transformed numbers of HIV-1 RNA copies per ml of plasma, excluding measurements obtained in the first 6 months after seroconversion and during advanced immunosuppression (i.e., with <100 CD4+ T cells per ml of blood). Consequently, 698 study participants were eligible for VL analysis.

## Human genotype data

DNA samples were genotyped in the context of previous GWAS (*Fellay et al., 2009*; *Pereyra et al., 2010*) or for the current study on various platforms, including the HumanHap550, Human 660W-Quad, Human1M and HumanOmniExpress BeadChips (Illumina Inc., San Diego, CA, USA), as well as the Genome-Wide Human SNP Array 6.0 (Affymetrix Inc., Santa Clara, CA, USA) (*Table 2*). Study participants were filtered on the basis of genotyping quality, a sex check, and cryptic relatedness. SNP quality control was performed separately for each dataset: SNPs were filtered on the basis of missingness (excluded if called in <99% of participants), minor allele frequency (excluded if <0.01), and marked deviation from Hardy-Weinberg equilibrium (excluded if p<0.00005). Missing genotype imputation was performed with the Mach software per genotyping platform (in separate batches for Illumina 1M, OmniExpress, 550K and Affymetrix data) using 1000 Genomes Phase I CEU population data as reference haplotypes. Imputed markers were filtered on minor allele frequency (excluded if <0.01) and imputation quality using Mach's reported r-squared measure (excluded if <0.3). SNPs with a deviation in the allele frequencies between platforms were excluded. High-resolution HLA class I typing (4 digits; HLA-A, HLA-B, and HLA-C) was obtained using sequence-based methods, or imputed from the SNP genotyping data as described elsewhere (*Jia et al., 2013*).

## HIV-1 sequence data

Near full-length retroviral sequence data were obtained by bulk sequencing of viral RNA present in pretreatment-stored plasma, and in 11 cases, of proviral DNA isolated from peripheral blood mononuclear cells, as previously described (*Sandonís et al., 2009*; *John et al., 2010*). We defined an amino acid residue as variable if at least 10 study samples presented an alternative allele. Per position, separate binary variables were generated for each alternate amino acid, indicating the presence or absence of that allele in a given sample.

## Association analyses

To globally assess the association between human genomic variation (SNPs), HIV-1 proteomic variation (amino acids) and clinical outcome (VL), we

**Table 2.** Distribution of samples across genotyping platforms and cohorts

| N | Genotyping platform | Cohort |
|---|---|---|
| 140 | Illumina 1M | ACTG |
| 6 | Illumina OmniExpress 12v1H | CARLOS III |
| 518 | Affymetrix 6.0 | HOMER |
| 136 | Illumina OmniExpress12v1H | HOMER |
| 47 | Illumina 650k | IHCS |
| 6 | Illumina 660W-Quad | IRSICAIXA |
| 2 | Illumina 1M | SHCS |
| 79 | Illumina 550k | SHCS |
| 122 | Illumina OmniExpress12v1H | SHCS |
| 15 | Illumina 550k | WAHCS |

ACTG = AIDS Clinical Trials Group Network; CARLOS III = Instituto de Salud Carlos III; HOMER = HAART Observational Medical Evaluation and Research Study; IHCS = International HIV Controllers Study; IRSICAIXA = AIDS Research Institute IrsiCaixa; SHCS = Swiss HIV Cohort Study; WAHCS = Western Australian HIV Cohort Study.

performed three series of analyses (*Figure 1*): [A] human SNPs vs VL; [B] human SNPs vs HIV-1 amino acids; and [C] HIV-1 amino acids vs VL. To test for association between human SNPs and HIV-1 amino acids, we used phylogenetically corrected logistic regression (*Carlson et al., 2008*; *Carlson et al., 2012*). For association testing between polymorphic amino acids in human HLA genes and HIV sequence variation, we used standard logistic regression (for a binary HLA amino acid) or a multivariate omnibus test (when more than one alternate allele was present) including sex, cohort, and the coordinates of the first two principal component axes as covariates. We used linear regression models in PLINK to test for association between human SNPs and VL, and between HIV-1 amino acids and VL (*Purcell et al., 2007*), including sex, cohort, and the coordinates of the first two principal component axes as covariates (*Price et al., 2006*). An additive genetic model was used for all analyses involving human SNPs. Significance was assessed using Bonferroni correction (significance thresholds of $7.25 \times 10^{-9}$, $2.4 \times 10^{-12}$, and $1.6 \times 10^{-5}$ for analyses A, B, and C, respectively, *Figure 1*).

## Acknowledgements

We would like to thank all the patients participating in these genetic studies, the many study nurses, physicians, data managers and laboratories involved in all the cohorts; Tanja Stadler and Sebastian Bonhoeffer (at ETH Zürich, Switzerland) and Samuel Alizon (at MIGEVEC, Montpellier, France) for helpful discussions; and Jennifer Troyer (at the Laboratory for Genomic Diversity, NCI) for her work on the HOMER genotyping data.

## Additional information

### Funding

| Funder | Grant reference number | Author |
|---|---|---|
| Swiss National Science Foundation | 33CS30_134277/Swiss HIV Cohort Study, 31003A_132863/1, PP00P3_133703/1 | Amalio Telenti, Jacques Fellay |
| Santos Suarez Foundation, Lausanne | | Amalio Telenti, Jacques Fellay |
| Hungarian Academy of Sciences | Bolyai János Research Fellowship | Viktor Müller |
| Michael Smith Foundation for Health Research | | Zabrina L Brumme |
| Canadian Institutes of Health Research | | Zabrina L Brumme |
| Sciex-NMS Program | 10.267 | István Bartha |
| Spanish Ministry of Science and Innovation | SAF 2007-61036, 2010-17226 and 2010-18917 | Cecilio López-Galíndez, Concepción Casado |
| Fundacion para la investigacion y prevencion del SIDA en Espana | 36558/06, 36641/07, 36779/08, 360766/09 | Cecilio López-Galíndez, Concepción Casado |
| RETIC de Investigacion en SIDA | RD06/006/0036 | Cecilio López-Galíndez, Concepción Casado |
| National Institute of Allergy and Infectious Diseases (NIAID) | P01-AI074415 | Todd M Allen |
| Bill and Melinda Gates Foundation | | Todd M Allen |
| SNF Professorship | PP00P3_133703/1 | Jacques Fellay |

The funders had no role in study design, data collection and interpretation, or the decision to submit the work for publication.

## Author contributions

IB, JMC, CJB, JL, NP, CL, NF, ZK, VM, Analysis and interpretation of data, Drafting or revising the article; PJM, AT, JF, Conception and design, Analysis and interpretation of data, Drafting or revising the article; ZLB, Acquisition of data, Analysis and interpretation of data, Drafting or revising the article, Contributed unpublished essential data or reagents; MJ, Acquisition of data, Drafting or revising the article, Contributed unpublished essential data or reagents; DWH, JM-P, JD, CL-G, CC, AR, HFG, EB, PV, TK, SY, SJO'B, PRH, Acquisition of data, Drafting or revising the article; TMA, Acquisition of data, Analysis and interpretation of data, Drafting or revising the article; DH, Analysis and interpretation of data, Drafting or revising the article, Contributed unpublished essential data or reagents

## Ethics

Human subjects: Participating centers provided local Institutional Review Board approval for genetic analysis. Study participants provided informed consent for genetic testing, with the exception of a subset where a procedure approved by the relevant Research Ethics Board allowed the use of anonymized historical specimens in the absence of a specific informed consent.

# Additional files

## Major dataset

The following datasets were generated:

| Author(s) | Year | Dataset title | Dataset ID and/or URL | Database, license, and accessibility information |
| --- | --- | --- | --- | --- |
| Bartha I, Carlson JM, Brumme CJ, McLaren PJ, Brumme ZL, John M, et al. | 2013 | Interactive HIV-Host Genome-to-Genome Map | http://dx.doi.org/ 10.5281/zenodo.7138 | Publicly available at Zenodo (https://zenodo. org). |
| Bartha I, Carlson JM, Brumme CJ, McLaren PJ, Brumme ZL, John M, et al. | 2013 | Online Supplementary Dataset of the HIV Genome-to-Genome Study | http://dx.doi.org/ 10.5281/zenodo.7139 | Publicly available at Zenodo (https://zenodo. org). |

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
