## [Decision Letter]

Thank you for sending your work entitled “A genome-to-genome analysis of associations between human genetic variation, HIV-1 sequence diversity, and viral control” for consideration at *eLife*. Your article has been favorably evaluated by a Senior editor, a Reviewing editor, and 3 reviewers.

The Reviewing editor and the reviewers discussed their comments before we reached this decision, and the Reviewing editor has assembled the following comments to help you prepare a revised submission.

The study represents an important and novel direction in the analysis of host-pathogen interactions, namely the joint analysis of both host and pathogen genomes. With additional information on pathogen phenotype (viral load) this enables the authors to provide a detailed dissection of how genetic variation within the players influences outcome.

Overall, we felt the study was well designed and analysed. Although there are perhaps no great surprises in terms of findings, the finding of many associated viral variants outside epitopes and the stronger association between human variation and HIV variation, compared to viral load, are both interesting results. When revising, we would like you to focus on the following issues:

1) The authors wish to include a clinical correlate in the study, and we understand the use of viral load as it is easy to measure and is associated with progression. However, viral load is not only the only marker of disease progression – CD4 count, rate of CD4 decline, immune activation, time to starting therapy from seroconversion etc, so it is clear that clinical progression is multifactorial. (It is also widely reported that CD8 immune responses, especially ELISpots, do not associate with viral load). With this in mind, the authors can only conclude that when using viral load as the surrogate their method produces stronger P values, but it cannot be stated that their ‘intermediate phenotype’ can replace possible other clinical correlates.

2) The authors have used sequences from proviral DNA and plasma DNA in the same analysis, without mentioning this apart from in the Methods. Whereas proviral DNA may represent a record of HLA-imposed selection pressure it may not represent the circulating virus, and therefore associations with viral load etc. may be misleading. One would expect to see some justification of this approach.

3) It would help frame the findings better if the power of the association studies was given. How big an effect size were the studies powered to detect? Given the finding of Alizon et al. cited here and other related papers, are we to be surprised by the lack of associations in the viral proteome to VL study? Does this place an upper bound for effect size of associations? Similarly for the lack of associations outside of MHC, which seem quite definitive especially in the case of the host genome to viral genome study.

4) The GWAS to viral sequence variation is probably the most interesting finding of this study, with 48 viral amino acids showing significant associations with host SNPs in the MHC. The strongest association is observed for position 135/Nef within an A*24:04 restricted epitope and for a SNP which is known to tag A*24:02. Assuming that this effect has a structural basis (i.e., the molecular interaction between peptide and MHC), why only show amino acids in the viral genome? Why not make it more “symmetric” and also look at the individual amino acids in the HLA molecules? Because the authors have imputed amino acid polymorphisms of the HLA proteins (Jia et al.), this should be relatively easy and potentially interesting.

---

## [Author Response]

*1) The authors wish to include a clinical correlate in the study, and we understand the use of viral load as it is easy to measure and is associated with progression. However, viral load is not only the only marker of disease progression – CD4 count, rate of CD4 decline, immune activation, time to starting therapy from seroconversion etc., so it is clear that clinical progression is multifactorial. (It is also widely reported that CD8 immune responses, especially ELISpots, do not associate with viral load). With this in mind, the authors can only conclude that when using viral load as the surrogate their method produces stronger P values, but it cannot be stated that their ‘intermediate phenotype’ can replace possible other clinical correlates*.

We fully agree that plasma viral load is one clinical variable associated with progression, but that it does not capture all aspects of clinical disease. We have modified the Abstract and the main text accordingly to correctly convey the message that the improvement in power refers to our greater capacity to detect host factors relevant to viral biology.

*2) The authors have used sequences from proviral DNA and plasma DNA in the same analysis, without mentioning this apart from in the Methods. Whereas proviral DNA may represent a record of HLA-imposed selection pressure it may not represent the circulating virus, and therefore associations with viral load etc. may be misleading. One would expect to see some justification of this approach*.

HIV-1 sequences were generated from proviral DNA in a very small number of study participants (N=11). In addition, there is no clear evidence that proviral DNA is less likely to reflect intra-host host pressure. While it has been shown that escape mutations in proviral DNA can lag a few months vs the circulating virus, in this cross-sectional study of chronically infected subjects most escape mutations should already have occurred and any minor delays in escape rates would not be expected to effect the results given the stability of viral loads during chronic infection.

*3) It would help frame the findings better if the power of the association studies was given. How big an effect size were the studies powered to detect? Given the finding of Alizon et al. cited here and other related papers, are we to be surprised by the lack of associations in the viral proteome to VL study? Does this place an upper bound for effect size of associations? Similarly for the lack of associations outside of MHC, which seem quite definitive especially in the case of the host genome to viral genome study*.

We agree with the reviewers that power calculations are important to frame our findings in their appropriate context and we have included these in the text of the manuscript.

*4) The GWAS to viral sequence variation is probably the most interesting finding of this study, with 48 viral amino acids showing significant associations with host SNPs in the MHC. The strongest association is observed for position 135/Nef within an A*24:04 restricted epitope and for a SNP which is known to tag A*24:02. Assuming that this effect has a structural basis (i.e., the molecular interaction between peptide and MHC), why only show amino acids in the viral genome? Why not make it more “symmetric” and also look at the individual amino acids in the HLA molecules? Because the authors have imputed amino acid polymorphisms of the HLA proteins (Jia et al.), this should be relatively easy and potentially interesting*.

We now include the analysis proposed by the reviewers. The amino acid association testing is largely consistent with the analysis of classical HLA alleles. It also adds a mechanistic dimension. We have added this analysis to the text and provide full association results online (http://g2g.labtelenti.org).